# Probiotics as Potential Therapeutic Agents: Safeguarding Skeletal Muscle against Alcohol-Induced Damage through the Gut–Liver–Muscle Axis

**DOI:** 10.3390/biomedicines12020382

**Published:** 2024-02-07

**Authors:** Martina Sausa, Alberto Fucarino, Letizia Paladino, Francesco Paolo Zummo, Antonio Fabbrizio, Valentina Di Felice, Francesca Rappa, Rosario Barone, Antonella Marino Gammazza, Filippo Macaluso

**Affiliations:** 1Department of Theoretical and Applied Sciences, eCampus University, 22060 Novedrate, Italy; martina.sausa@gmail.com (M.S.); alberto.fucarino@uniecampus.it (A.F.); antonio.fabbrizio@uniecampus.it (A.F.); 2Department of Biomedicine, Neurosciences and Advanced Diagnostics, University of Palermo, 90127 Palermo, Italy; letizia.paladino@unipa.it (L.P.); francescopaolo.zummo01@unipa.it (F.P.Z.); valentina.difelice@unipa.it (V.D.F.); francesca.rappa@unipa.it (F.R.); rosario.barone@unipa.it (R.B.); antonella.marinogammazza@unipa.it (A.M.G.); 3Euro-Mediterranean Institute of Science and Technology (IEMEST), 90139 Palermo, Italy

**Keywords:** alcohol, skeletal muscle, probiotics, intestinal microbiota, axis gut–muscle, axis liver–muscle

## Abstract

Probiotics have shown the potential to counteract the loss of muscle mass, reduce physical fatigue, and mitigate inflammatory response following intense exercise, although the mechanisms by which they work are not very clear. The objective of this review is to describe the main harmful effects of alcohol on skeletal muscle and to provide important strategies based on the use of probiotics. The excessive consumption of alcohol is a worldwide problem and has been shown to be crucial in the progression of alcoholic liver disease (ALD), for which, to date, the only therapy available is lifestyle modification, including cessation of drinking. In ALD, alcohol contributes significantly to the loss of skeletal muscle, and also to changes in the intestinal microbiota, which are the basis for a series of problems related to the onset of sarcopenia. Some of the main effects of alcohol on the skeletal muscle are described in this review, with particular emphasis on the “gut-liver-muscle axis”, which seems to be the primary cause of a series of muscle dysfunctions related to the onset of ALD. The modulation of the intestinal microbiota through probiotics utilization has appeared to be crucial in mitigating the muscle damage induced by the high amounts of alcohol consumed.

## 1. Introduction

Excessive alcohol consumption ranks as the third leading cause of health impairment globally, contributing to 5.3 percent of all yearly deaths. About 43% of the population over the age of 15 years has consumed alcohol in the past 12 months, indicating a risk of death and disability in early childhood due to this cause [1]. Despite the significant social, economic, and medical burdens of alcohol consumption on individuals and countries, alcohol consumption has increased worldwide in recent decades, especially during the COVID-19 pandemic [2]. Importantly, there are different ranges of alcohol consumption levels between men and women [3]. For men, moderate consumption is defined as two drinks per day, while fifteen or more per week constitutes excessive consumption. For women, however, one drink per day defines moderate consumption, while eight or more per week reflects excessive consumption [4].

Alcohol consumption, especially heavy consumption over time, has been shown to be crucial in the etiology and progression of alcoholic liver disease (ALD), which encompasses a broad spectrum of liver diseases ranging from hepatitis to steatosis, fibrosis, cirrhosis, and liver neoplasia [5]. ALD often leads to malnutrition, a prevalent complication, although currently effective therapies for its reversal are lacking. This term, “malnutrition”, is broad, with its major component being sarcopenia or the loss of skeletal muscle, causing significant adverse effects in those with liver disease [6]. In ALD, alcohol and its metabolites contribute to skeletal muscle loss by hindering protein synthesis and causing anabolic resistance, thereby reducing the effectiveness of nutritional interventions. Although the consequences of alcohol in skeletal muscle may regress after complete abstinence, recovery is usually incomplete and may be related to underlying liver disease [7,8].

Among the many factors contributing to the pathogenesis of alcohol-induced liver damage, bacterial products of intestinal origin appear to play a central role in inducing steatosis and inflammation. Specifically, elevated levels of lipopolysaccharide (LPS) are found in the blood of patients with chronic alcohol consumption, and this phenomenon is related to a number of factors, including changes in the composition of the gut microbiota (dysbiosis) [9]. Normally, ethanol is absorbed from the gastrointestinal tract through simple diffusion. While a small quantity of ethanol is metabolized in the stomach during the first pass, prolonged and excessive alcohol consumption can alter the composition of intestinal bacteria, making the intestines more permeable and leading to the activation of systemic inflammatory cascade pathways [10].

Numerous nutritional approaches aim to enhance muscle mass or mitigate muscle loss. These methods encompass protein and micronutrient supplementation, effectively curbing inflammation and oxidative stress. Additionally, certain strategies leverage the gut microbiota, which is believed to play a role in the onset of various metabolic disorders. Indeed, alterations in the microbiota have demonstrated correlations with obesity, diabetes, and cardiovascular disease—conditions typically linked to the emergence of insulin resistance and inflammation. Altered gut microbiota composition, impaired physiological status, and muscle catabolic states, suggest that the microbiota, directly or indirectly, could influence the condition and regulation of muscle mass. The hypothesis of a “gut-muscle axis” has been proposed by several authors [11,12,13], i.e., the impact of the gut microbiota and its integration with the host gut on skeletal muscle metabolism and function. Recent studies have developed nutritional strategies that include probiotics in controlling muscle mass and function. Probiotics, live microorganisms beneficial to the health when consumed in sufficient quantities, have traditionally been lactic acid bacteria (LAB) and bifidobacteria strains derived from humans or food sources. However, current approaches emphasize leveraging bacteria naturally existing in the gut. The mechanisms through which probiotics operate are intricate and have not been fully comprehended, but one of their objectives involves adjusting the composition of disrupted gut microbiota, such as in individuals with high alcohol intake.

This narrative review describes the system-wide complications of alcohol abuse, with particular emphasis on the effects of ethanol on skeletal muscle. Specifically, the role of the gut microbiota and the function of the intestinal barrier, liver, and systematic inflammation in the “gut-liver-muscle axis” are discussed, with the aim of understanding how the alteration of these factors affects muscle mass and function, thereby establishing a connection with sarcopenia, either directly or indirectly. Finally, various nutritional strategies to prevent and/or treat alcohol-induced sarcopenia are presented. 

The present study acquired scientific articles for analysis via the built-in search engine on PubMed. The search was based on the following keywords: (“Alcohol,” or “Ethanol,”) and (“Skeletal muscle,”); (“Probiotic,”) and (“Skeletal muscle,”); (“Alcohol,” or “Ethanol,”) and (“Intestinal Microbiota,” or “Gut Microbiota,”); (“Axis Gut-Muscle,”); (“Axis Liver-Muscle”). The articles, published in the last 10 years, recovered by the search engine were subjected to a detailed examination to identify the damage of alcohol consumption on skeletal muscle and intestinal microbiota and the effects of probiotics on the same organs. The analysis of each clinical study included the examination of the effects of alcohol on skeletal muscle and the effects of probiotics on the liver and skeletal muscle, taking into account the sex, age, and health status of the participants (if they were healthy or had a particular alcohol-related disease), including the ethanol dosage and the type/dose of the probiotics administered.

## 2. Malnutrition and Sarcopenia Associated with Alcoholic Liver Disease

Malnutrition is a common complication of ALD and causes sarcopenia (Figure 1), characterized by a generalized loss of skeletal muscle mass and strength [14]. Several factors contribute to skeletal muscle mass regulation, including protein turnover, cellular energy status, availability of metabolic substrates, endocrine disruption, cytokines, myostatin, and exercise [15]. However, the direct effects of ethanol and its metabolites on skeletal muscle, along with the consequences of liver disease, cause disturbances in proteostasis (protein homeostasis) and subsequent sarcopenia. Once ALD evolves into cirrhosis, alcohol abstinence is unlikely to completely reverse sarcopenia [7], as other factors, including hyperammonemia, hormonal, and cytokine abnormalities, exacerbate this condition and maintain a state of ethanol-initiated anabolic resistance [16].

Although sarcopenia is a debilitating process most often related to aging [17], characterized by frailty, disability, and increased risk of fractures related to falls, in younger individuals, sarcopenia can result from a variety of factors, including metabolic syndrome, physical inactivity, inadequate nutrition, alterations in the gut microbiota, neuromuscular diseases, and high alcohol consumption [18]. For this reason, an important distinction is made between primary sarcopenia, or loss of muscle mass and strength that accompanies aging, and secondary sarcopenia, which occurs in disease states [19].

Another commonly used term is alcoholic myopathy (Figure 1), which often refers to muscle damage rather than muscle loss. Although skeletal muscle injury can occur during acute alcohol abuse, probably due to the direct effects of alcohol or its toxic metabolite, acetaldehyde, a much more frequent consequence is skeletal muscle loss due to concomitant alcohol-related liver damage [20]. Most of these studies were conducted in animal models, in which the duration of treatment and levels of alcohol consumption were defined by the researchers. From these experimental studies, it is apparent that chronic alcohol consumption leads to reduced cross-sectional area (CSA) of type 2 fibers and protein synthesis, activation of pro-inflammatory cytokines, hyperammonemia, muscle autophagy, and mitochondrial alterations [15,16,21,22,23]. On the other hand, there is a great difficulty in conducting the same studies in humans, where it often becomes complicated to identify the concentration of alcohol to be administered, as well as the duration of treatment, ultimately risking unreliable results.

### 2.1. Malnutrition

Malnutrition is a common feature among patients with ALD and is also an important prognostic factor of the disease. Energy intake is usually reduced in these patients due to several factors: loss of appetite, altered taste, alcohol dependence, and physical inactivity [24]. Malnutrition is a condition that can lead to changes in both physical and mental function, negatively impacting one’s quality of life and clinical outcomes [25].

The ESPEN Consensus Statement and the Academy of Nutrition and Dietetics (Academy) have outlined diagnostic criteria for malnutrition in adults [26,27].

Therefore, malnutrition can be diagnosed today if there is either of the following exist:(a)Reduction in body mass index (BMI) < 18.5 kg/m^2^ or underweight as defined by the World Health Organization (WHO);(b)Combined weight loss and BMI reduction (with age-dependent cut-off);(c)Sex-dependent reduction in fat mass index (FFMI);(d)A combination of the six defined parameters (low energy intake, weight loss, muscle mass loss, subcutaneous fat loss, fluid accumulation, and hand grip strength), at least two of which are met.

Malnutrition in patients with cirrhosis is associated with increased complications requiring hospitalization, ultimately leading to death [28]. Most of the basic tools used to assess nutritional status are not reliable in cirrhosis [29]. Liver failure results in decreased rates of albumin, pre-albumin, retinol-binding protein, and transferrin. BMI and weight are not reliable measures of fluid overload. This assessment is important considering that half of patients with cirrhosis awaiting liver transplantation (LT) are registered for decompensated disease [30]. Consequently, malnutrition in cirrhosis is very difficult to assess. Despite the significant consequences of malnutrition in patients with ALD, nutritional assessment is not often performed even in specialized hepatology centers. Therefore, there is a strong need for dedicated tools including specific criteria and cut-off points.

### 2.2. Sarcopenia

The exploration of sarcopenia in individuals with a history of high alcohol consumption spans several years. This condition is characterized by a gradual and extensive decline in muscle mass, strength, and functionality [31]. In contrast to the geriatric literature that encompasses both muscle mass and function in defining sarcopenia, studies on cirrhosis predominantly focus on defining sarcopenia as the loss of muscle mass. Different methods, including anthropometry, bioelectrical impedance analysis (BIA), ultrasound, MRI (Magnetic Resonance Imaging), and CT (Computer Tomography) scans, have been used to measure muscle mass in decompensated cirrhosis [32]. The European Working Group on Sarcopenia in Older People (EWGSOP) introduced a new three-step definition of sarcopenia, marking a significant shift from the previous definition, which solely focused on diminished muscle mass [33,34]. This updated definition now incorporates muscle function, signifying a noteworthy change in approach. In these updated guidelines, there is a notable emphasis on muscle strength, acknowledging its superior predictive ability for adverse sarcopenia outcomes compared to muscle mass [35]. Additionally, sarcopenia compromises muscle quality, a term encompassing both microscopic and macroscopic aspects of muscle structure and composition. However, due to technological limitations, muscle mass and quality cannot be considered primary parameters to define sarcopenia, but muscle strength must also be added [36]. Therefore, based on these three parameters, EWGSOP defines sarcopenia using three criteria:(1)Criterion 1: Low muscle strength;(2)Criterion 2: Low muscle quantity or quality;(3)Criterion 3: Low physical performance.

When diminished muscle strength is identified, sarcopenia becomes a likely consideration. The diagnosis hinges on the presence of reduced muscle mass or compromised quality. Severe sarcopenia is established when low strength, inadequate muscle quantity/quality, and diminished physical performance are all evident. Sarcopenia’s presence often contributes to complications such as infections, hepatic encephalopathy, and ascites, and it is linked to decreased overall survival [37]. In addition, mortality increases for sarcopenic patients who are waiting for liver transplantation compared to non-sarcopenic patients [38]. Very often, sarcopenia is strongly considered to be equivalent to malnutrition, as other causes such as immobilization, endocrine disorders, or neurological disorders can lead to the onset of sarcopenia. However, today, there are important guidelines for classifying malnutrition and sarcopenia, and this could be very important in determining specific nutritional tools for patients with ALD.

## 3. Gut Microbiota and Hypothesis of “Gut-Liver-Muscle Axis” in ALD

The gut microbiota refers to the collection of microbial organisms, including bacteria and other microbes, residing in the gastrointestinal tract. Comprising tens of trillions of microorganisms, the predominant groups belong to the phyla *Firmicutes* and *Bacteroides* [39]. These microbial communities play crucial roles in various aspects of human health and physiology. The intestinal microbiota plays a crucial role in different mechanisms: the maturation and continuous stimulation of the immune response of the host [40]; the maintenance of the integrity of the intestinal barrier, limiting the perpetuation of pathogens in the intestine [41]; the modulation of host cell proliferation [42]; vascularization [43]; the regulation of intestinal, neurological, and endocrine functions and bone density regulation [44,45].

In a healthy colon, the gut microbiota and the host maintain a mutually beneficial and symbiotic relationship. This microbial community demonstrates remarkable adaptability, swiftly restoring microbial balance (eubiosis) following an acute disturbance or insult. This ability to recover and maintain equilibrium is pivotal for overall gut health and function. In the case of high alcohol consumption or in the presence of liver pathologies, instead, changes to the normal composition of the intestinal microbiota occur instead, which are indicated by the term dysbiosis and can have harmful effects on the host. 

High alcohol consumption, then, influences what is called the “gut-liver axis”, a highly relevant mechanism for ALD progression [46]. Overall, the structure of the liver is connected intricately with the intestine, where nutrients and the microbiome play a crucial role in sustaining a well-functioning metabolism and a healthy liver. Nutrients originating from the intestine travel to the liver via the portal circulation. The gradual blood flow within the hepatic sinusoids facilitates interactions among substances from the intestine, hepatocytes, other cells in the liver parenchyma, and hepatic immune cells. This interaction is enhanced by the fenestrated endothelium present in the sinusoids [47]. As the largest immune organ, the liver accommodates a diverse range of immune cells, and it possesses a remarkable capability to attract and activate immune cells in response to signals derived from metabolic changes or pathogens originating in the gut. LPSs originating from Gram-negative bacteria in the intestinal microflora typically penetrate the mucosa in minimal quantities. They enter the portal circulation and undergo elimination in the liver, playing a crucial role in regulating immune homeostasis. Resident macrophages, known as Kupffer cells, and hepatocytes jointly participate in this process through distinct LPS recognition systems [48]. The scientific community has a keen interest in understanding the impact of the gut microbiota on liver disease. A recent study highlighted the pivotal role of the liver in mediating the interaction between the host and the gut microbiota [49]. It is noteworthy that bile acids, generated by the liver, can exert a modulatory effect on the microbiome, as certain bacteria utilize bile acids in their metabolic processes [50]. The interplay between the microbiome and the host liver is particularly intriguing in ALD. Alcohol has been demonstrated to alter the microbiome composition, compromising intestinal integrity and barrier function, making this interaction a focal point of interest in ALD research [51].

The integrity of the intestinal mucosa relies on the functioning of various components, including a protective layer of defensins on the intraluminal surface of the intestinal epithelium, tight junction (TJ) proteins connecting intestinal epithelial cells, and the presence of immune cells within the intestinal wall. Alcohol exerts both direct and indirect effects on these functions in the gut. The direct effects involve its impact on the functions of the gut components, such as the defensins and TJ proteins. Indirect effects occur as a result of alcohol and/or its metabolites being distributed through the bloodstream, influencing the overall integrity and function of the intestinal mucosa [52]. Moreover, elevated blood alcohol levels have been linked to a decrease in the expression of mRNA levels for crucial proteins involved in the formation of TJ proteins between colonic epithelial cells. In Caco-2 intestinal epithelial cells, it has been observed that alcohol leads to a reduction in the expression of TJ proteins, specifically occludin and zona occludens-1 (ZO-1) [53].

The interaction of the components belonging to the “gut-liver axis” determines in fact the behavior of different mechanisms that are a part of it, such as the composition of the microbiota, the function of the intestinal barrier, liver, and systemic inflammation, all severely altered mechanisms in the ALD [10]. Factors contributing to dysbiosis in ALD are not fully understood; however, environmental and genetic factors, increased gastric pH, intestinal dysfunction, etc., participate in dysbiosis and its development [54]. Two more studied factors contributing to intestinal dysbiosis in ALD concern the down-regulation of intestinal antimicrobial peptides (AMPs) and the accumulation of bacterial products in the portal circulation. AMPs normally play a role in the innate defense of the host against bacteria and maintain homeostasis of the intestinal mucosa [55]; but in the case of chronic alcohol intake, there is a reduction in the expression of AMPs in the intestine, resulting in dysbiosis, reduction of intestinal barrier, and systemic inflammation [56]. On the other hand, chronic alcohol ingestion can lead to excessive intestinal bacterial proliferation, with an increase in the serum level of bacterial products, including LPSs or endotoxins that interact with Toll-like receptors (TLRs) present in different types of cells, including Kupffer cells and hepatocytes, resulting in increased intestinal permeability and the onset of inflammatory processes [57].

Indeed, various compounds generated or influenced by the gut microbiota have the potential to enter the systemic circulation, thereby influencing skeletal muscle cells. This intricate relationship, termed the “gut-liver-muscle axis” and represented in Figure 2, outlines how the gut microbiota can impact muscle mass, quality, and function through the mediating liver. The compounds involved in this interplay can significantly contribute to the modulation of skeletal muscle health and performance. The healthy intestinal microbiota produces folate and vitamin B12, which improve muscle anabolism and prevent oxidative stress induced by hyperhomocysteinemia and endothelial damage, leading to a reduction in muscle function [58,59]. The gut microbiota plays a role in synthesizing essential amino acids, among them tryptophan, which serves as crucial substrates for muscle protein anabolism [60]. Tryptophan, in particular, has the capacity to stimulate the IGF-1/p70s6k/mTor pathway within muscle cells, thereby encouraging the expression of genes responsible for myofibril synthesis [61]. This pathway activation contributes significantly to the process of building muscle fibers. Liver diseases, including ALD and end-stage liver disease, frequently involve associated muscular alterations that contribute to a poorer clinical prognosis. Recent studies have shed light on the detrimental effects of these muscle changes on liver function, giving rise to the hypothesis of a bidirectional relationship known in the literature as the “liver-muscle axis”.

## 4. Effect of Alcohol on Skeletal Muscle

Ethanol metabolism primarily occurs in the liver and brain; however, emerging evidence indicates that its metabolism also takes place in skeletal muscle. Distinguishing the clinical and pathophysiological effects stemming from underlying liver disease and its repercussions from the direct impact of ethanol (or its metabolites) on skeletal muscle proteostasis can pose challenges. Furthermore, the combined detrimental effects of liver disease and ethanol/metabolites might contribute to the development of sarcopenia [8]. This section discusses some of the effects that ethanol has on skeletal muscle (see Table 1), providing an overview of the molecular pathways involved in regulating mass, metabolism, and/or function.

### 4.1. Hyperammonemia

Several studies have shown that hyperammonemia is commonly present in patients with ALD and plays a major role in the onset of sarcopenia in these patients and in individuals who engage in high alcohol consumption [68]. Indeed, in liver disease, the liver’s capacity to detoxify ammonia is compromised. In response, skeletal muscle often assumes a compensatory role in the metabolism and clearance of ammonia [69].

It is known that a high ammonia concentration in skeletal muscle alters protein metabolism through reduced protein synthesis and increased autophagy [68]. In fact, in vitro experiments have shown that elevated ammonia concentration activates myostatin through the nuclear factor kappa-light-chain-enhancer (NF-kB)-dependent pathway (Figure 3), which, in turn, down-regulates mTOR and increases the phosphorylation of adenosine monophosphate-activated protein kinase-alpha2 (AMPK-α2), resulting in reduced protein synthesis and increased autophagy proteolysis [70].

The high concentration of muscle ammonia also stimulates the metabolism of glutamine and glutamate into α-ketoglutarate and ammonia by the enzyme glutamate dehydrogenase (GDH) (Figure 3) [71]. Under these conditions, the flux of the production of tricarboxylic acid intermediates (TCAs) decreases to prevent the accumulation of anions within the mitochondrial matrix. As a consequence, reduced mitochondrial function results in reduced adenosine triphosphate (ATP) synthesis and, ultimately, protein synthesis [70]. All of this, in turn, can cause oxidative damage to lipids and proteins, further exacerbating sarcopenia [72]. Furthermore, in hyperammonemic states, protein synthesis is impaired due to the increased phosphorylation of eukaryotic initiation factor 2 (eIF2), an important factor involved in translation initiation [70].

### 4.2. Direct Effects of Ethanol and Its Metabolites

Several studies have demonstrated increased autophagy and proteostasis in the skeletal muscle of patients with ALD, resulting in sarcopenia [73]. It appears that enzymes that metabolize alcohol, particularly aldehyde dehydrogenase (ALDH) inhibitors, contribute to these events [8]. Most of the ethanol in the body is metabolized in the liver by the enzyme alcohol dehydrogenase (ADH), which converts ethanol into a toxic compound called acetaldehyde. However, the latter generally has a short half-life; in fact, it is rapidly broken down into acetate by ALDH. Eventually, acetate is broken down into carbon dioxide and water in other tissues.

In the skeletal muscle of ALD patients, the concentration of ethanol is remarkably high, and due to the inhibition of ALDH, there is an increase in the concentration of the metabolite acetaldehyde, which, despite having a short half-life before being converted to acetate, causes numerous damages observed in these patients (Figure 3). In ALD patients, acetaldehyde impairs ornithine transcarbamylase, an enzyme that plays an essential role in the urea cycle, whose main purpose is to capture toxic ammonia and convert it into urea, a less toxic source of nitrogen for the body [74]. Consequently, muscle loss in these subjects could be the result of the combined effect of ethanol and ammonia within skeletal muscle.

Animal models fed ethanol show impaired protein synthesis [75], while patients with alcoholic cirrhosis show alterations in protein synthesis in the post-absorptive and postprandial phases, suggesting a state of anabolic resistance, i.e., with the same amount of protein consumption in the diet, protein formation is less active [76]. Indeed, in ALD, there is an alteration of the proteosome-ubiquitin pathway (UPP) and an increase in autophagic processes, which collectively contribute to the significant loss of muscle mass. Overall, therefore, ethanol and its metabolites cause metabolic, biochemical, and molecular perturbations in skeletal muscle.

### 4.3. Endocrine Abnormality

The effects of high alcohol consumption have also been associated with endocrine abnormalities. In general, the endocrine system is a complex system of glands that produce and secrete hormones directly into the blood circulation, with actions prolonged over time. Substance abuse, such as chronic alcohol consumption, has been shown to have serious adverse effects on several components of the endocrine system in both women and men. Alcohol activates the hypothalamic-pituitary-adrenal (HPA) axis, which results in an increase in the concentration of adrenocorticotropic hormone (ACTH) and glucocorticoids [77].

Excessive alcohol consumption and alcoholism have been linked to impaired reproductive function in both men and women. The proper functioning of the reproductive system relies on the hypothalamic–pituitary–gonadal (HPG) axis and its associated hormones. Studies have indicated that individuals with a history of heavy alcohol consumption often experience HPG dysfunction, resulting in diminished libido, infertility, and gonadal atrophy. The dysregulation of the HPG axis not only contributes to reproductive issues but also poses risks of other significant health problems, including mood and memory disorders, osteoporosis, and muscle atrophy [78]. Specifically, in men, alcohol abuse can negatively affect testosterone production, leading to alcoholic hypogonadism. This results in reduced libido, erectile dysfunction, and decreased fertility. Furthermore, it can alter sperm development, compromising reproductive health. In women, however, alcohol can disturb the hypothalamic–pituitary–ovarian (HPO) axis, influencing the production of estrogen and progesterone. This can lead to menstrual irregularities, anovulation, and infertility. Furthermore, excessive alcohol consumption during pregnancy is associated with serious consequences for fetal development [79]. In human studies, alcohol abuse demonstrates multiple effects on lowering testosterone levels, with these effects contingent on the quantity and duration of alcohol consumption. Chronic intake consistently leads to decreased testosterone levels [66]. However, some studies suggest that heightened alcohol consumption might not have a direct correlation with testosterone levels [80]. In contrast, other studies indicate that alcohol intake is positively correlated with testosterone level [81].

An important relationship also exists between testosterone and cortisol based on the hypothalamic–pituitary adrenal and gonadal axes; in this context, ethanol increases cortisol levels and reduces testosterone levels [82]. The consequences of these hormonal changes are different but very important: cortisol stimulates protein breakdown in muscle tissue, while low testosterone levels can hinder muscle growth, especially in men [83,84].

The decreased secretion of IGF and disruptions in circadian patterns can also play a role in muscle loss. IGF has various impacts on skeletal muscle, promoting protein synthesis and muscle growth through IGF induction and myostatin inhibition [85]. These mechanisms collectively stimulate muscle growth and maintenance. Testosterone also contributes to myostatin inhibition, so it is presumed that the reduction of these hormones in ALD patients contributes to impaired protein synthesis and increased myostatin [86]. Myostatin plays a crucial role in proteostasis by inhibiting protein synthesis and, consequently, leading to a reduction in muscle mass. In individuals with cirrhosis, the serum myostatin level can be up to four times higher than that observed in healthy subjects [87]. In individuals with ALD, myostatin not only hinders skeletal muscle protein synthesis but also triggers protein degradation through the ubiquitin–proteasome system (UPS) [62]. In mouse models, the suppression of myostatin expression has been associated with positive outcomes, including enhancements in skeletal muscle mass, hepatic insulin sensitivity, and reductions in liver fat [88]. Significant studies have underscored the role of myostatin in liver fibrosis, attributing its influence to interactions with hepatic stellate cells and the promotion of systemic inflammation. This, in turn, leads to the expression of interleukins, including transforming growth factor-beta1 (TGF-β1) [89].

Finally, ethanol exerts profound effects on calcium homeostasis and vitamin D metabolism. Low levels of 25 hydroxyvitamin D3 (25(OH)D_3_) and/or low levels of 1,25 dihydroxyvitamin D3 (1,25(OH)_2_) have been found in alcoholic subjects and ethanol-treated animals [90]. In addition to possible effects on reduced vitamin D intake or synthesis related to the direct and indirect effects of ethanol and/or lifestyle in ALD patients, ethanol has recently been shown to alter the renal production of 1,25(OH)_2_, affecting both the synthesis and metabolic inactivation of 1,25(OH)_2_ [91]. This effect could be related to the increase in ethanol-induced oxidative damage causing to a reduction in the plasma levels of 1,25(OH)_2_.

### 4.4. Myosteatosis

Extremely common among ALD patients is myosteatosis, characterized by excessive fat accumulation within skeletal muscle leading to an imbalance between lean and fat muscle mass, eventually contributing to the decline in muscle function [16]. There are three potential fat deposition phenotypes: (1) intermuscular adipose tissue (IMAT), (2) intramuscular adipose tissue, and (3) intramyocellular lipid droplets (IMCLs) [92]. It preferentially affects certain groups of muscles on the basis of their oxidative capacity. In fact, oxidative muscles combat IMCL accumulation by increasing the level of β-oxidation and producing free fatty acids (FFAs) [93]. In contrast, glycolytic muscles face increased intramyocellular triglyceride (TG) accumulation due to the re-esterification of FFAs as a consequence of reduced mitochondrial oxidative phosphorylation [94].

Physiologically, it has been proposed that IMCL acts as an intracellular energy source during exercise. Indeed, the content of IMCL decreases during prolonged exercise, and, akin to glycogen, it increases in a trained state. It has long been acknowledged that during resistance exercise, fat oxidation plays a significant role in meeting the energy demands of skeletal muscle. While fatty acids (FAs) stored in adipose tissue as TG must undergo lipolysis, be released into the bloodstream, and be transported to active muscles for oxidation, IMCL reserves present a readily available substrate source during resistance exercise. A significant study has suggested that not all fat oxidation during exercise may be attributed to plasma FAs oxidation, indicating that lipid droplets within muscle cells play a crucial role as a fuel source during exercise. Particularly in prolonged exercise, IMCLs are believed to serve as a significant fuel source, and the oxidation of IMCLs might have a preserving effect on glycogen oxidation [95].

In ALD patients, alterations in lipid metabolism and mitochondrial dysfunction potentially contribute to myosteatosis. Specifically, lipid accumulation within muscle fibers over time leads to muscle fiber atrophy [96]. Numerous studies have established a robust association between alcohol intake and insulin resistance. Alcohol consumption appears to significantly impede the usual metabolic reactions of skeletal muscle in response to insulin (Figure 3) [97]. Normally, insulin stimulates the movement of glucose from the bloodstream to various tissues, including skeletal muscle, via a transporter known as glucose transporter 4 (GLUT4). Insulin resistance disrupts the signaling pathway of insulin. Faulty signaling obstructs the uptake of excess blood glucose by the muscle due to impaired GLUT4 translocation [98]. Consequently, insulin resistance is linked to changes in lipid metabolism, typically characterized by heightened levels of circulating FAs and TGs, along with increased accumulation of lipid intermediates like diacylglycerols (DG), ceramides, and long-chain coenzyme A (LC-CoA) [99].

In normal circumstances, insulin triggers the activation of phosphatidylinositol 3-kinase (PI3K) and other downstream intermediates, including protein kinase B (Akt), to exert its effects on skeletal muscle. The activation of these intermediates leads to enhanced glucose transport via GLUT4, moving from intracellular vesicles to the cell’s outer membrane, and promotes glycogen synthesis by activating glycogen synthase [100]. In patients with ALD or those abusing alcohol, the accumulation of lipids adversely impacts the initiation of insulin signaling pathways (Figure 3). 

### 4.5. Gut Dysfunction

Changes in the gut microbiota and the breakdown of epithelial cell TJs induced by excessive ethanol consumption can lead to disruptions in circulating cytokine levels and increased presence of LPSs [101,102,103]. These alterations have direct implications for skeletal muscle proteostasis and can contribute to the onset of sarcopenia through various mechanisms [104]. These aspects are discussed in more detail in Section 5, going on to analyze the possible role of the gut microbiota on the “gut-liver-muscle axis” and how possible nutritional strategies can reduce the deleterious effects caused by ethanol in ALD patients.

## 5. Effects of Alcohol on the Gut Microbiota 

### 5.1. Alteration of the Composition of the Intestinal Microbiota

Recently, numerous experimental and clinical studies have highlighted the direct correlation between alcohol intake and alterations in the composition of the intestinal microbiota. Alcohol and its metabolites exert direct or indirect influences on the gut microbiota, leading to changes in its composition. This can occur through alcohol’s ability to either inhibit or promote the proliferation of intestinal bacteria, often by modifying the microenvironment within the gut [105]. It is interesting to note that while alcohol exposure does not necessarily affect the overall amount of intestinal microbiota, it does significantly alter its composition. In murine models, alcohol intake was found to decrease the relative abundance of *Lactobacillus* (or *Sporalactobacillus*) while increasing the relative abundance of *Allobaculum* [106]. These shifts in specific microbial populations highlight the impact of alcohol on the intricate balance within the gut microbiota.

Continued alcohol consumption also increases the relative abundance of the genera *Ruminococcus* and *Coprococcus*, inducing liver damage and endotoxemia [107]. In human models, other alterations of the intestinal microbiota have been found, and they have always been related to excessive alcohol consumption or to pathologies including ALD. One is the abundance of the relative genus of the phylum *Proteobacteria* and the genera *Sutterella*, *Holdemania*, and *Clostridium* in excessive-alcohol consumers [108]. There is also a relative abundance of phyla *Proteobacteria* and *Firmicutes* and of the class *Gammaproteobacteria*, and a reduction of phylum *Bacteroides* and the classes *Clostridia*, *Bacteroidetes*, and *Verrucomicrobiae* in patients with ALD [109].

Evidently, the gut microbiota’s alterations often result in an increased proportion of Gram-negative bacteria, forming the foundation for inflammation induced by LPSs. In a depleted microbiota scenario, *Firmicutes* represent Gram-positive bacteria, while *Bacteroides* are Gram-negative bacteria, typically considered beneficial or innocuous. This shift in the balance of Gram-negative and Gram-positive bacterial populations can significantly impact gut health and the inflammatory response within the body.

### 5.2. Disfunction of the Intestinal Barrier

Under physiological conditions, the intestinal barrier is composed of several layers [110,111]: the outer one includes the mucus layer, the commensal intestinal microbiota, and defense proteins such as AMPs and secretory immunoglobulin A (sIgA). Intestinal epithelial cells (IEC) constitute the intermediate layer, while the inner part is composed of immune cells of innate and adaptive immunity [112]. The intestinal barrier acts as a crucial defense, effectively segregating microorganisms within the intestinal lumen from entering the bloodstream. Simultaneously, it facilitates the passage of nutrients from the lumen into the portal vein, ensuring a valuable and non-toxic blood supply for the body [113]. This selective permeability is pivotal for maintaining a balance between protection from harmful substances and absorption of essential nutrients. In patients with ALD or in subjects with high alcohol consumption, there is a breakdown of the intestinal barrier, and the main mechanisms are related to alcohol and ADH, the alteration of the motility of the small intestine, changes in gastric secretion, and increased LPSs from enterobacteria [114]. Alcohol and ADH can lead to mucus erosion and ulceration within the gastrointestinal tract. Additionally, they have the capacity to modify the glycosylation of the protective mucus layer, potentially resulting in increased intestinal permeability [115]. These effects can compromise the integrity of the gut barrier, contributing to a heightened permeability that allows for substances to pass through the intestinal lining more easily, potentially leading to inflammation and other health issues. When alcohol is ingested, it can be absorbed into the duodenum and fasting; at this point, it can be metabolized in the intestinal barrier or continue to spread into the circulation to be transported to various body districts [116]. It is important to note that ADH levels are significantly elevated in the intestine following the administration of large amounts of alcohol, compared to acetate [117]. Chronic alcohol consumption prompts an elevation in the microsomal ethanol-oxidizing system through cytochrome P450 (CYP) enzymes, particularly CYP 2E1. Initially, CYP 2E1 facilitates the ethanol oxidation to acetaldehyde and further metabolizes it into acetate. This enzymatic process is a crucial step in the breakdown of ethanol within the body, playing a significant role in alcohol metabolism. The catalytic reaction of ethanol mediated by CYP 2E1 generates significant ROS, such as anion superoxide, hydrogen peroxide, and hydroxyl radical, which can induce direct damage to liver cells’ inflammation and oxidative stress [118].

### 5.3. Role of Short Chain Fatty Acids (SCFAs)

The most studied mediator regarding the effect of the gut microbiota on skeletal muscle function is SCFAs [119]. The gut–muscle axis describes how the gut microbiota can impact muscle mass, muscle quality, and muscle function. Recently, some studies have focused their attention on the role of SCFAs, which play an important role in modulating lipid, carbohydrate, and protein metabolism in skeletal muscle (Figure 3). Although SCFAs are formed in the gut, effective concentrations can be found circulating in the body [120]. SCFAs are formed from the fermentation of fibers such as non-digestible carbohydrates, and they include acetate, propionate, and butyrate. These SCFAs are critical for maintaining the integrity of the epithelial barrier, the loss of which compromises barrier permeability and increases the risk of bacteria or bacterial antigen translocation. This event triggers the inflammatory cascade which may underpin the chronic inflammation observed in obesity and insulin resistance [121]. 

SCFAs primarily target the mitochondria within skeletal muscle cells [122]. These compounds can enter the systemic circulation and are absorbed by skeletal muscle cells, where they act as ligands for free fatty acid receptors 2 and 3 (FFAR-2 and FFAR-3) [123]. These receptors play a pivotal role in modulating glucose absorption and metabolism, and in promoting insulin sensitivity [122]. Moreover, SCFAs contribute to the up-regulation of the NAD-dependent deacetylase sirtuin-1 (SIRT1) receptor, which serves as a regulator of mitochondrial biogenesis. Notably, the expression of mitochondrial proteins correlates positively with the abundance of SCFAs produced in the intestines of individuals with inflammatory bowel disease, highlighting a strong link between the microbiota and mitochondrial function [124]. This connection underscores the impact of gut microbial products on mitochondrial health and function. SCFA supplementation can restore/improve muscle mass and/or decrease strength due to antibiotic treatment [125].

### 5.4. Inflammation

It has been shown that an altered composition of the microbial ecosystem may be associated with an imbalance between anti-inflammatory and pro-inflammatory intestinal responses leading to low-grade systemic chronic inflammation [126]. High alcohol consumption can trigger immune system dysregulation, reducing the body’s capacity to combat the colonization of pathogenic bacteria within the intestines. This impairment, along with intestinal barrier dysfunction, contributes to “intestinal leakage” or increased intestinal permeability, resulting in systemic inflammation [113]. This cascade of events leads to elevated levels of circulating endotoxins such as LPSs, exacerbating the inflammatory response throughout the body. Precisely, LPSs traverse into the bloodstream and interact with specific receptors, notably TLR-4 present on innate immune cells. When LPS binds to TLR-4, it initiates the recruitment of diverse intracellular components, such as mitogen-activated protein kinase (MAPK), setting off cellular signaling cascades. One of the consequential pathways activated by this interaction is the pro-inflammatory pathway NF-kB [127]. This series of events triggers an inflammatory response within the body, perpetuating the immune reaction. It has been found that NF-kB is involved in muscle atrophy, participating in the degradation of muscle proteins through the UPP, inducing inflammation and, consequently, blocking the regeneration of muscle fibers [128]. In addition to TLR-4, other inflammatory factors include tumor necrosis factor alpha (TNF-α), interleukin-1 (IL-1), interleukin-8 (IL-8), interleukin-10 (IL-10), and interferon gamma (IFN-γ) [129]. In skeletal muscle, TNF-α activates the expression of NF-kB-related genes that reduce cell differentiation and proliferation (via myogenin and MyoD inhibition) [130]. Elevated levels of circulating LPSs stimulate the heightened expression and production of pro-inflammatory cytokines by immune cells. This inflammatory response influences several metabolic pathways, impacting protein homeostasis and mitochondrial function, which collectively contribute to muscle atrophy. The regulation of these pathways by the pro-inflammatory cytokines plays a significant role in driving the process of muscle atrophy. In the case of acute inflammation, the stimulation of muscle proteolysis (UPP) is observed, a reduction in protein synthesis (via mTor-regulated initiation of translation), and also the induction of cellular apoptosis and/or the inhibition of satellite stem cell differentiation [131].

Evidently, oxidative stress induced by ROS plays a significant role in promoting inflammation by triggering inflammatory signaling pathways and further enhancing ROS production, establishing a detrimental cycle. High alcohol consumption contributes to the increased expression of cyclooxygenase 2 (COX-2), inducible nitric oxide synthase, and transient activation of REDOX-sensitive transcription factors like NF-kB [132]. Ethanol administration decreases the level of the NF-kB inhibitor while promoting its localization, thereby fostering the expression of inducible nitric oxide synthase (iNOS) and nitric oxide (NO) production, thus promoting inflammation [133]. Ethanol metabolism through CYP 2E1 initiates the formation of ROS. Alcohol exposure enhances CYP 2E1 expression while inhibiting antioxidant enzyme expression and cellular protective molecules, thereby facilitating ROS production [134]. ROS further stimulates the TLR-4 signaling cascade, culminating in the activation of NF-kB and the release of inflammatory factors, particularly TNF-α [105]. This cascade of events significantly contributes to the inflammatory response associated with alcohol consumption.

## 6. Modulation of the Gut Microbiota—Benefits of Probiotics

The gut microbiota is arguably the most flexible human organ. Achieving eubiosis is the goal of the therapeutic modulation of the intestine, both through targeted and non-targeted approaches. The former includes modulation by diet, antibiotics, prebiotics, and probiotics; the latter, instead, provides for the use of bacterial metabolites and the host as a target [135].

The WHO has defined probiotics as “living microorganisms that if administered in adequate doses provide benefits to the health of the host” [115]. Nobel laureate Elie Metchnikoff introduced the concept of probiotics to the scientific community. He published a seminal report linking the longevity of Bulgarians with the consumption of fermented milk products containing viable *Lactobacilli* [136]. This observation suggested that certain microbes, when ingested, could be beneficial for human health. Since then, probiotics have been widely marketed and consumed, mostly as dietary supplements or functional foods. Probiotics modulate the gut microbiota, promoting an anti-inflammatory environment that counteracts bacterial translocation and endotoxin production, and improves the integrity of the intestinal barrier [137]. Some of the benefits of probiotics are depicted in Figure 4. Numerous articles, reviews, and systematic reviews have highlighted the impact of probiotics on host health. These sources extensively discuss the preventive role of probiotics in various health issues, encompassing conditions such as inflammatory bowel disease (IBD), antibiotic-associated diarrhea, infection-induced diarrhea, as well as allergic rhinitis and allergic disorders like atopic dermatitis (eczema) [138,139,140,141].

While numerous probiotic strains are extensively acknowledged as safe, and some are officially designated as “generally recognized as safe”, both the European Food Safety Authority (EFSA) and the US Food and Drug Administration (FDA) refrain from attributing the capacity to prevent or treat diseases to the administration of probiotics. In various countries, probiotics are acquired as dietary supplements and adhere to prevailing market regulations. The EFSA rejected requests for approval primarily due to several key reasons, including insufficient characterization, a lack of pertinent human studies, absence of measurable outcomes demonstrating direct human benefits, and concerns about the quality of the presented studies [142]. Similar to the EFSA, the FDA has not granted approval for any probiotics in the prevention or treatment of health issues. Both regulatory agencies emphasized the need for probiotics to be tailored to the health needs of specific individuals. They highlighted the importance of comprehending the rationale behind applied principles and stressed that numerous studies conducted by researchers and companies are essential to gain a thorough understanding of the potential mechanism of action of a specific probiotic, which is often not extensively elucidated [143]. In contrast, Health Canada has granted approval for a multi-strain probiotic and a single strain as a natural health product specifically intended for a relief of symptoms associated with IBS [144].

Over the last two decades, there has been a significant increase in both the quantity and quality of clinical studies examining the health advantages associated with probiotics. Similar to any intervention, it is crucial not only to assess the benefits but also to understand the associated risks. In the early stages, probiotics were linked to traditional uses in naturally fermented food products and were, therefore, not classified as drugs. This historical classification may have contributed to a lack of emphasis in earlier probiotic re-search on monitoring and reporting adverse events. Notably, within the general population and among non-immunocompromised or severely debilitated patients, acute safety concerns seem to be relatively minimal, especially given the widespread global utilization of probiotics in foods and nutritional supplements [145]. Furthermore, recent clinical trials exhibit significantly enhanced reporting of adverse events. However, similar to most interventions, long-term safety endpoints are infrequently tracked by investigators. Given that probiotics are live products, there are theoretical risks associated with potential long-term impacts on the microbiota, immune system, cardiometabolic functions, and other physiological parameters. These aspects warrant further discussion and investigation.

A particularly relevant aspect to take into consideration is the diversity of the effects of probiotics in different populations by age, gender, and health condition. In this regard, long-term studies aimed at demonstrating the safety of probiotics in populations at risk, including the elderly, newborns, individuals with weakened/compromised immune function, are scarce. While positive effects of probiotics have been documented in various groups, it is important to note that immune-compromised individuals may face an elevated risk of adverse events [146]. The potential risks of probiotics for pregnant and breastfeeding women have also been investigated. Out of 100 studies, only 11 reported adverse events that could potentially be linked to the treatment, encompassing issues such as gastrointestinal problems, nausea, and headaches. Notably, no serious health problems for either the mother or child were reported in these studies [147]. Lastly, a category warranting consideration in the context of probiotic effects is that of newborns, particularly preterm infants who present a significant opportunity for the modulation of microbiota structure and function [148]. Therapies directed at the neonatal microbiota have the potential to impact host biology throughout the lifespan, introducing foreign microbial strains when conditions are more conducive to colonization or influencing the early developmental trajectories of crucial organs, including the brain [149]. As of now, there is limited evidence suggesting that probiotic supplementation in early life has a negative impact on neurodevelopmental outcomes. Interestingly, there was a notable reduction in the cases of deafness observed in children treated with probiotics [150].

In summary, addressing potential long-term concerns associated with probiotics is challenging due to limited available data. Ongoing research in this field is crucial to un-cover new insights into long-term safety implications that must be considered in safety assessments. Additionally, further studies are required to elucidate which specific high-risk groups necessitate closer and more prolonged follow-up for a comprehensive understanding of the safety profile. 

### Mechanisms of Actions of Probiotics

Traditionally, probiotics available on the market predominantly consisted of LAB and bifidobacteria strains derived from human or food sources. However, recent strategies are shifting focus towards utilizing bacteria that naturally reside in the intestine. This approach aims to harness the potential benefits of the existing gut microbiota for improved health outcomes. The rationale is to introduce or enhance the growth of specific bacterial strains already present in the gut to positively influence overall gut health and function. However, additional research is essential to evaluate the utility and long-term safety of probiotics across diverse disease conditions. The advantages of probiotics are particularly evident in diseases associated with the gastrointestinal tract [151]. Although probiotics are generally considered safe, occasional side effects may occur, including constipation, flatulence, hiccups, nausea, infections, and rashes.

The precise mechanisms underlying the actions of probiotics are complex and have not yet been fully elucidated. However, one of their primary purposes involves modulating the composition of an altered intestinal microbiota, which is particularly relevant in individuals who engage in high alcohol consumption. An imbalance in the gut microbiome, coupled with a reduction in the population of bacteria producing metabolites like SCFAs, is frequently observed in individuals with conditions such as IBD, type 2 diabetes (T2D), obesity, autoimmune diseases, and among cancer patients [152]. 

Probiotics function as antimicrobial agents by producing various substances such as SCFAs, organic acids, hydrogen peroxide, and bacteriocins. This activity contributes to the reduction of pathogenic bacteria in the gut, as observed in a study by Fantinato et al. [153] with *Streptococcus salivarius*. Furthermore, probiotics enhance intestinal barrier function by promoting the production of mucin proteins and regulating the expression of TJ proteins. This includes the modulation of occludin, ZO-1, and claudin-1, which collectively contribute to the maintenance of a robust and effective intestinal barrier [154]. Therefore, by influencing the microbial community in the gut, probiotics strive to promote a healthier gut environment and contribute to overall well-being, especially in populations with imbalances due to alcohol consumption. 

Another mechanism of action employed by probiotics is competitive exclusion. This phenomenon occurs when bacterial species sharing the same ecological niche engage in competition for limited resources such as nutrients and space. Competitive exclusion involves two primary strategies: (1) exploitation competition, and (2) interference competition. The first is an indirect mechanism characterized by the rapid consumption of resources, thereby limiting the availability of competing organisms and promoting the probiotic’s own growth. The second mechanism, however, occurs when one organism directly harms another, often through the production of antimicrobial compounds as an example. This direct interference aims to gain a competitive advantage in the ecological niche [155]. Through the production of antimicrobial agents and metabolic compounds, probiotics are able to suppress the growth of other microorganisms [156] as well as compete for receptors and binding sites with other intestinal microbes on the intestinal mucosa [157]. Among the most studied strains, *Lactobacillus* has been shown to improve the integrity of the intestinal barrier, which may in turn lead to the maintenance of immune tolerance and the decrease in the translocation of bacteria through the intestinal mucosa [158]. 

In parallel, probiotics exert systemic effects, which include the modulation of immune response. They are able to interact with the intestinal immune system, contributing to a more balanced immune response and reducing systemic inflammation. The intestinal immune system comprises physical barriers, such as the epithelium and the underlying connective tissue known as the lamina propria, housing immune effector cells. The gut-associated lymphoid tissue (GALT) plays a crucial role in immune functions and serves as a significant source of T and B cells. Probiotics actively participate in both innate and adaptive immune responses by influencing various immune cells, including dendritic cells (DCs), macrophages, and B and T lymphocytes. Interactions between probiotics and the host intestinal cells predominantly occur at the surface of the intestinal barrier. Upon consumption, probiotic bacteria adhere to intestinal epithelial cells, activating them through pattern recognition receptors (PRRs). This binding prompts the release of cytokines, leading to the activation of regulatory T cells, which help maintain immune homeostasis in the intestinal mucosa. DCs play a pivotal role in processing intestinal antigens. They can activate naïve CD8+/CD4+ T cells and direct helper T (Th) cell responses toward Th1, Th2, and Th17, resulting in the production of (IFN)-γ, (IL)-4, IL-5, and IL-17. Additionally, probiotics induce the maturation of B cells into immunoglobulin IgA-producing plasma cells. These plasma cells migrate across the epithelium into the mucus layer, where they regulate bacterial adhesion to host tissue. Probiotics also exhibit anti-inflammatory effects by down-regulating TLRs expression, secreting metabolites that inhibit TNF-α entry into blood mononuclear cells, and inhibiting NF-κB signaling in enterocytes, contributing to the suppression of intestinal inflammation [155]. 

In addition, probiotics can directly affect nutrient uptake and metabolite production. Some studies suggest that probiotics can improve the absorption of nutrients in the intestinal tract, thus contributing to the maintenance of metabolic health. These effects can extend beyond the intestine, positively affecting the metabolism and overall health of the body [159]. 

## 7. Role of Probiotics in Ethanol-Induced Muscle Damage

The positive effects of probiotics are not limited to the gastrointestinal tract and the liver; they extend also to muscle tissue, offering interesting prospects to improve the damage induced by ethanol at the systemic level (Table 2). In recent years, the use of probiotics has demonstrated an innovative ability to mitigate systemic inflammation, offering promising prospects for muscle health [160]. Studies have shown that probiotics, known for their positive impact on intestinal health, can have beneficial effects on the systemic level, helping to reduce inflammation with significant impacts on muscle tissue. Skeletal muscle, essential for mobility and motor function, is significantly affected by systemic inflammatory processes. In this context, probiotics, acting as modulators of the inflammatory response, could interfere with the pro-inflammatory pathways, reducing the production of inflammatory cytokines and excessive immune response, phenomena often linked to muscle damage. Moreover, the positive effect of probiotics on the balance of intestinal flora can have a direct impact on muscle health. A balanced gut microbiota can positively influence the inflammatory response at the systemic level, reducing the risk of chronic inflammation that can compromise muscle tissue. Using probiotics as modulators of systemic inflammation could protect skeletal muscle from inflammatory damage and promote its optimal functionality.

As previously written, one of the common complications observed in ALD patients or subjects who consume high amounts of alcohol is sarcopenia. Many studies have explored cross-talk between the intestine and muscle, finding that the composition and environmental interaction of bacterial flora can affect the quality, function, and energy metabolism of muscles [129]. Based on these studies, it is understood how the modulation of what has been called the “gut-liver-muscle axis” can have a positive effect on the health of skeletal muscle damaged by ethanol. One of the interactions between intestinal microbiota and muscles concerns the metabolites of the microbiota itself, which can be involved in stimulating the energy metabolism of muscles and improving the performance of resistance. Acetic acid, for instance, is recognized for its beneficial effects. It has been found to enhance glucose absorption and fatty acid metabolism by AMPK. Additionally, acetic acid increases the expression of GLUT4 and myoglobin through the myocyte enhancer factor 2A (MEF2A) pathway [161]. This metabolic pathway plays a significant role in inducing an increase in muscle mass, thus contributing to its potential to promote muscle growth and metabolic health. The close correlation between skeletal muscle and intestinal flora is an interesting aspect to the possible use of probiotics in the mitigation of damage caused by the excessive consumption of alcohol. The harmful effects of alcohol on the balance of intestinal flora can compromise muscle function and health, but the strategic use of probiotics would offer an innovative solution to mitigate these damages. It has been widely shown that alcohol causes significant imbalances in the intestinal flora, generating unfavorable conditions such as harmful excessive bacterial growth and chronic inflammation. This compromised intestinal environment can adversely affect the health of the skeletal muscle since the systematic inflammation and insufficient absorption of nutrients can impair its functionality. Probiotics, acting as promoters of intestinal health, are able to restore the balance of intestinal flora impaired by alcohol. These beneficial bacteria can compete with pathogens, reducing the excess of harmful bacteria and restoring an optimal intestinal environment. This can promote a reduced inflammatory response and better nutrient absorption, directly contributing to the preservation of skeletal muscle health.

Another interesting aspect concerns the regulation of myokine function by SCFAs [162]. Myokines are released by myocytes and are involved in the metabolism of muscle and other tissues (adipocytes, liver, brain) through their receptors. Several myokine notes, including myostatin, irisin, mionectin, fibroblast growth factor 21 (FGF-21), and interleukin- 6 (IL-6), have not been well characterized, as have their functions. However, some characteristics of these molecules have been identified, including myocyte proliferation, differentiation, muscle growth, and atrophy. While irisin, mionectin, and FGF-21 have a positive effect on muscle mass gain, myostatin and IL-6 are involved in muscle atrophy [163]. High alcohol consumption interferes with the production and secretion of these molecules, affecting communication between muscle and other tissues. This alteration can have a negative impact on the metabolic and inflammatory response of the skeletal muscle, compromising its function and adaptability. The use of probiotics would, therefore, restore the balance of myokines, thus preserving cell communication and the health of skeletal muscle compromised by alcohol. 

An important consequence of excessive alcohol consumption concerns the depletion of the muscle glycogen reserve. Glycogen content in both the liver and muscle tissue is a key factor in determining aerobic energy metabolism. The reduction of glycogen can, in fact, lead to an insufficient energy intake as well as the reduction in strength and muscle function, with the consequent impairment of energy metabolism [164]. Recently, it has been observed that probiotics (*Lactobacillus acidophilus*) regulate genes related to glycogen synthesis, such as glycogen synthase kinase-3 beta (GSK-3β) and Akt, and the glycogen content in tissues [165]. Optimizing and increasing glycogen levels through the modulation of the intestinal microbiota through the use of probiotics can effectively improve muscle function, thus delaying the deleterious effects found in ALD patients.

Probiotics can also significantly increase the oxidation of SCFAs and fatty acids and activate the peroxisome proliferator-activated receptor γ coactivator 1α (PGC-1α), increasing ATP production and providing the energy needed for proper skeletal muscle function [119]. Chronic alcohol abuse can destabilize the oxidation balance, leading to oxidative stress and negatively impacting muscle health; the increased production of free radicals disturbs the redox balance, compromising the skeletal muscle’s overall well-being. This oxidative stress damages muscle tissue, reducing its function and impairing recovery capacity. Probiotics, as modulators of the intestinal environment, can play a crucial role in restoring an optimal redox balance. By reducing oxidative stress and enhancing endogenous antioxidant defenses, probiotics have the potential to shield skeletal muscle from oxidative damage and encourage regeneration.

These research areas promise avenues for better understanding the role of probiotics in counteracting the muscle damage caused by alcohol abuse. However, it is crucial to acknowledge that many of these theories require further study and investigation to confirm their effectiveness and gain a complete understanding of the involved mechanisms.

**Table 2 biomedicines-12-00382-t002:** Studies investigating the effects of probiotics in liver and skeletal muscle.

Subjects (n)	Age	Physical Condition	Dosage	Effects	Reference
46 (men and women)	21–67 years	Moderate alcohol-associated hepatitis	*Lactobacillus rhamnosus GG* (LGG) for 6 months	Improvement in both liver injury and drinking	[166]
33 men and 39 women	23–63 years	Non-alcoholic fatty liver disease (NAFLD)	300 g daily of probiotic or conventional yogurt for 8 weeks	Reduction of serum levels of alanine aminotransferase, aspartate aminotransferase, total cholesterol, and low-density lipoprotein cholesterol	[167]
150 patients	?	Alcoholic liver injury	*Lactobaillus casei Shirota* (LcS) (low-dose, 100 mL, and high-dose, 200 mL) for 60 days	Improving lipid metabolism and regulating intestinal flora disorders	[168]
215 patients	?	Cirrhosis	*Probiotics VSL#3*, 112.5 billion CFU containing 8 strains of bacteria	Improving nutritional status and reducing severity of liver disease	[169]
66 men	?	Alcoholic psychosis	*Bifidobacterium bifidum* and *Lactobacillus plantarum 8PA3* for 5 days	Restoration of the bowel flora and great improvement in alcohol-induced liver injury	[170]
10 men	22.0 ± 2.4 years	Healthy resistance-trained	*Bacillus coagulans* BC30 + 20 g casein twice daily (500 M)	Increasing athletic performance, with an effect on peak power and fat mass	[171]
30 men	20–40 years	Healthy without professional exercise training	*Lactobacillus plantarum* TW10-HK (heat-killed TWK10) 3 × 10^10^ cells/day for 6 weeks.	Reduced physical fatigue and improved exercise endurance capacity and handrip strength	[172]
17 men	20.5 ± 0.8 years	Soldiers from an elite combat unit	*Bacillus coagulans* GBI-30 + Hydroxymethybutyrate calcium (CaHMB) 10^9^ CFU/day for a total of 40 days	Attenuated inflammatory response to intense military training and maintaining muscle integrity	[173]
20 men	18–30 years	Healthy, non-obese	*Streptococcus thermophilus*, *Lactobacillus acidophilus*, *Lactobacillus delbrueckii ssp. Bulgaricus*, *Lactobacillus paracasei*, *Lactobacillus plantarum*, *Bifidobacterium longum*, *Bifidobacterium infantis*, and *Bifisobacterium breve* + high-fat and high-energy diet two sachets (450 billion bacteria per sachet) for 4 weeks	Protection from body mass gain and fat accumulation	[174]
26 men and 12 women	20–40 years	Healthy	Live *Lactobacillus paracasei* (L-PS23) or heat-killed *Lactobacillus paracasei* (HK-PS23) 2 × 10^10^ cells/day for six weeks	Preventing strength loss after muscle damage and improving blood muscle damage and inflammatory markers, with protective, accelerated recovery and anti-fatigue benefits	[175]
18 (men and women)	19–26 years	Amateur runners	*Lactiplantibacillus plantarum* TWK10 10^10^ CFU/day for 6 weeks	Increasing muscle mass and endurance performance	[176]

## 8. Final Considerations

It is known that alcohol abuse causes a series of damages at a systemic level, including skeletal muscle. This review highlights how treatment with probiotics could be an excellent strategy to mitigate these damages. The effects of ethanol on some molecular pathways particularly important for the correct physiology of skeletal muscle are discussed; among these is the protein synthesis induced by myostatin, the mitochondrial biogenesis activated by SIRT-1 following the binding of SCFA with FFAR-2/3 receptors, as well as glycogen synthesis by insulin. These mechanisms are significantly inhibited by excessive alcohol abuse, leading to the onset of the phenomenon of sarcopenia, also observed in patients with ALD. This complication appears to be related to what is called the “gut-liver-muscle axis”. In fact, excessive alcohol consumption leads to alterations in the intestinal microbiota capable of triggering inflammatory processes that have repercussions on the entire organism, including the skeletal muscle. It, therefore, seems that modulating the microbiota through the use of probiotics is an excellent alternative for counteracting the muscle damage induced by excessive alcohol abuse. In conclusion, some studies have been conducted on the beneficial effects of the use of probiotics on the muscle. In fact, various scientific tests have shown how the consumption of probiotics can attenuate inflammatory responses, mitigate the loss of muscle mass, preventing muscle strength following muscle damage induced, for example, by intense physical effort. These studies could represent the point of connection between the benefits of probiotics and the damage caused to skeletal muscle by excessive alcohol consumption.

Any progress in this direction is, therefore, desperately needed, given the serious consequences of alcohol abuse. Therefore, focusing on the type or mix of probiotics to use to counteract muscle damage would be an excellent alternative for those patients with ALD or often hospitalized patients with serious muscle conditions. A valid strategy could be to focus on the mechanisms underlying the “gut-liver-muscle axis” in order to better understand this close connection at a systemic level and be able to reduce the serious consequences of sarcopenia. In this direction, further studies are needed to delve into the molecular mechanisms through which probiotics influence the muscle damage caused by alcohol. This could include studying the interactions between probiotics, intestinal microbiota, and molecular pathways involved in the inflammatory response and muscle homeostasis. Indeed, inflammation plays a fundamental role in the development and progression of pathologies linked to alcohol abuse, with serious repercussions also in skeletal muscle. Therefore, investigating the role of systemic and local inflammation in alcohol-induced muscle damage and understanding how probiotics can modulate this inflammatory response could be an important challenge to counteract the deleterious effects of alcohol abuse. This could include studies on cytokine production and the regulation of immune responses.

Finally, a last, no less important, aspect to take into consideration concerns clinical interventions. It would be useful to conduct large-scale clinical trials to evaluate the effectiveness of probiotics in preventing or attenuating alcohol-related muscle damage in humans. These studies should include different populations, probiotic dosages, and alcohol consumption regimens. This is related to the exploration of new strains of probiotics or the combination of different strains that could provide specific benefits for muscle health. The diversity of available probiotics could, indeed, offer personalized approaches for different individuals. From this point of view, it is also necessary to evaluate the long-term effects of taking probiotics on muscle condition and general health, especially in contexts of moderate or sporadic alcohol consumption, as well as examine the effects of probiotics on physical performance and the ability to recover after physical exercise, especially in the presence of alcohol consumption. Exploring these aspects could help to more precisely delineate the role and potential of probiotics in preventing or mitigating muscle damage associated with alcohol consumption.

Although the purpose of this review is to provide an overview of the effects of probiotics in the treatment of alcohol-induced muscle damage, there are some limitations that need to be considered. One of these is individual differences. Responses to interventions with probiotics can vary greatly between individuals due to differences in the composition of intestinal flora, metabolism, and other biological factors, such as simply sex, age, and health condition. Related to this aspect is the variability in the composition of probiotics. Several studies have analyzed different strains of probiotic bacteria, and the specific composition of probiotics can affect the results. Some studies may, in fact, use more effective strains than others, but, in the end, it is always necessary to evaluate well the condition of the individual. Another limitation of this review concerns the methodology of the study. Variability in study designs, including the number of participants, the duration of the study, and the inclusion and exclusion criteria, may influence the validity and generalizability of the results. Such studies are often aimed, for example, at people with serious diseases, and, therefore, it is difficult to respect the entire duration of the trial. In addition to this, the dose of probiotics and the duration of treatment may vary between studies, and the amount of probiotics can have a significant impact on results. Some studies may use too low doses or insufficient treatment periods. To this is added the lack of standardization in treatment protocols and measured results that can make it difficult to directly compare study results. Another limitation is publication bias. Unfortunately, studies showing positive results are more likely to be published than those showing no effect. This can lead to a distorted view of the effectiveness of probiotics in the treatment of alcohol-induced muscle damage. In addition, alcohol-induced muscle damage can be affected by many other factors, such as lifestyle, overall diet, and physical activity, which may not be fully controlled in studies. The interactions of probiotics with other factors must also be considered. In order to fully evaluate the effectiveness of probiotics in the treatment of alcohol-induced muscle damage, it is important to consider all the scientific evidence and pay attention to the methodological details of each study.

## Figures and Tables

**Figure 1 biomedicines-12-00382-f001:**
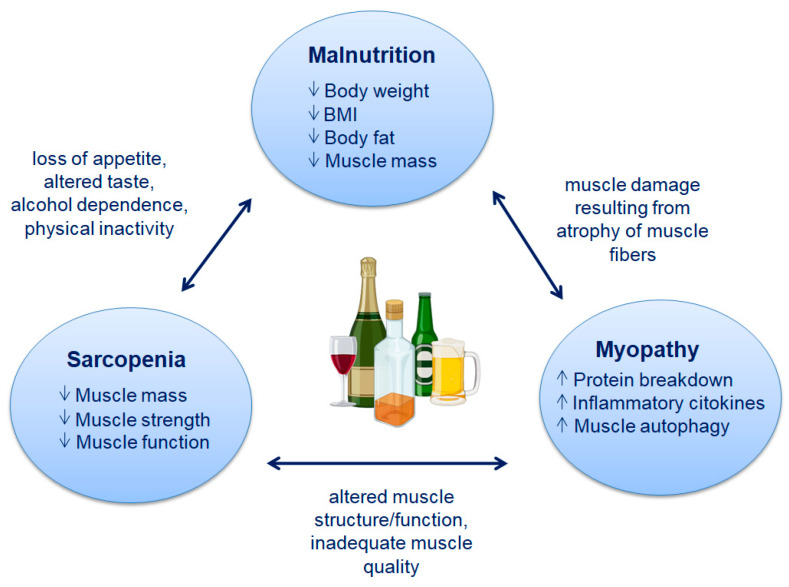
Characteristics of malnutrition, sarcopenia, and myopathy following alcohol abuse.

**Figure 2 biomedicines-12-00382-f002:**
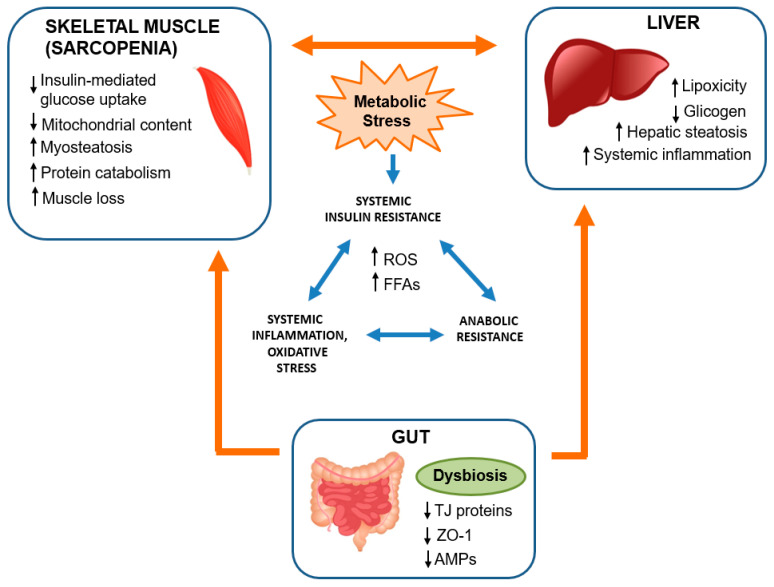
The interaction interplay among the gut, liver, and muscle involves several keys underlying mechanisms, many of which connect Alcoholic Liver Disease (ALD) to the development of sarcopenia [14,18]. Alcohol abuse initiates dysbiosis, leading to significant repercussions in the reduction of TJ proteins (such as ZO-1) and AMPs, which play a crucial role in bacterial defense and maintenance [57]. The destruction of the intestinal barrier results in an increase in bacterial products, including LPSs, which interact with hepatocytes, triggering inflammatory processes [9]. While the majority of ethanol metabolism occurs in the liver, in conditions of elevated concentration, as seen in ALD, various mechanisms lead to lipotoxicity and systemic inflammation characterized by an increase in ROS and FFAs. FFAs, transported though the bloodstream, accumulate in skeletal muscle, giving rise to the condition known as myosteatosis [16]. Currently, the inflammatory processes initiated by intestinal dysbiosis can activate the UPS system, leading to heightened protein catabolism and, consequently, reducing muscle growth, thus contributing to the onset of sarcopenia [62]. TJ: thin junction proteins; ZO-1: zona-occludens-1; AMPs: antimicrobial peptides; LPSs: lipopolysaccharides; ROS: reactive oxygen species; FFAs: free fatty acids.

**Figure 3 biomedicines-12-00382-f003:**
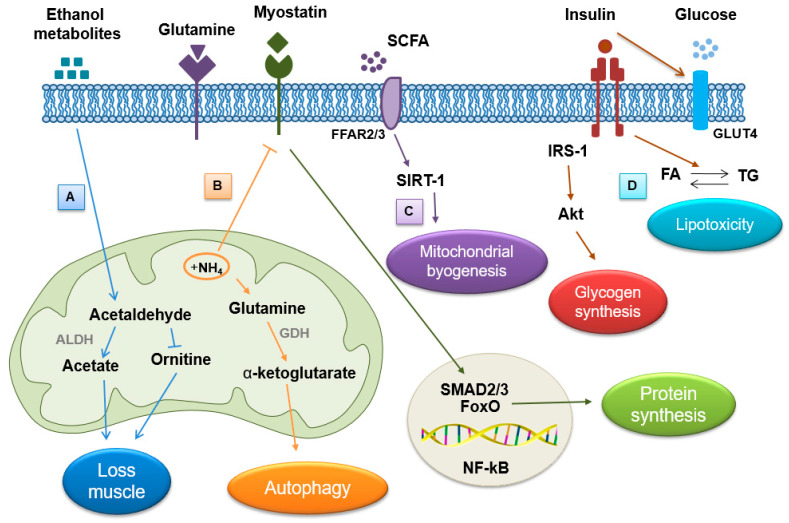
Schematic representation of the main effects of alcohol on skeletal muscle: (**A**) Under normal conditions in skeletal muscle cells, ethanol and its metabolites (acetaldehyde) are metabolized by ALDH to acetate, but high amounts of alcohol lead to the inhibition of ALDH, which, by altering the urea cycle, induces muscle loss. (**B**) Hyperammonemia activates, on the one hand, the autophagy process, stimulating the metabolism of glutamine into α-ketoglutarate, and, on the other hand, it inhibits myostatin, increasing muscle loss by reducing protein synthesis. (**C**) Ethanol interferes with mitochondrial biogenesis by reducing circulating SCFAs. (**D**) A condition of insulin resistance induced by excessive consumption of ethanol leads to alterations in lipid metabolism, characterized by an increase in circulating fatty acids and triglycerides, which induce lipotoxicity. ALDH: aldehyde dehydrogenase; SCFAs: short chain fatty acids.

**Figure 4 biomedicines-12-00382-f004:**
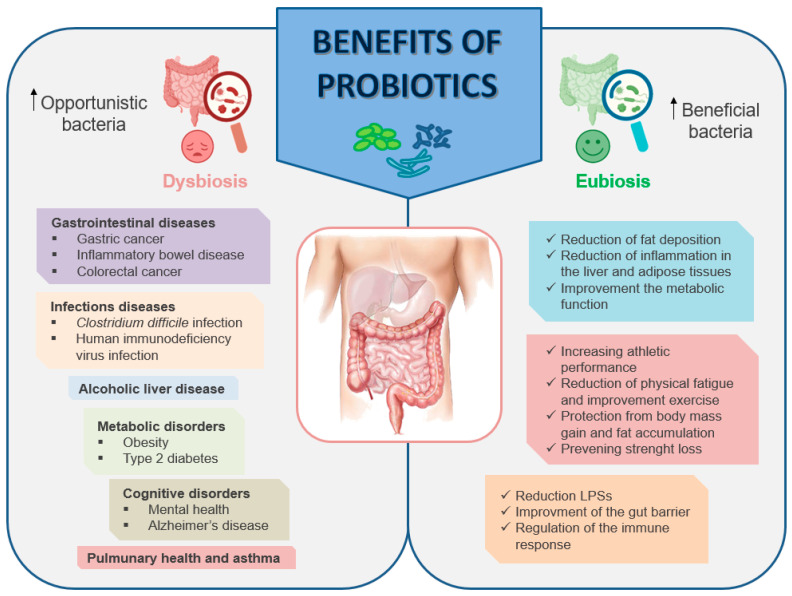
Schematic representation of the main benefits of probiotics.

**Table 1 biomedicines-12-00382-t001:** Studies investigating the effects of alcohol in skeletal muscle.

Subjects (n)	Age	Physical Condition	Dosage	Effects	Reference
19 men	18–40 years	Healthy moderate-drinking	100 mL whisky (32 g alcohol per day) or mineral water daily for 4 weeks	Increase in adiponectin concentration, and, in particular, HMW (high molecular weight) adiponectin, associated with skeletal muscle oxidative capacity	[63]
31 men and 4 women	53.3 ± 11.58 years	Alcoholic patients	>150 g alcohol/day during a prolonged period (>5 years), with an estimated total, lifelong consumption of 24 ± 16 kg ethanol/kg body weight	Increase in IL-15 correlated with increased protein content in muscle fibers, promoting myogenic differentiation and muscle growth	[64]
30 women	43 ± 5 years	Alcoholic patients hospitalized with a regular intake of alcohol of 3 years	During hospitalization (2–3 weeks), the patients abstained from alcohol, but before they consumed 4 units/day ethanol and 14 units/week ethanol	Atrophy both fast and slow muscle fibers, impairments in IGF-1 dependent signaling and pathways controlling translation initiation (AMPK/mTor/4E-BP1), increase in level of calpain-1 and ubiquitinated proteins	[65]
46 men	29.6 ± 4.2 years	Alcohol abusers	3–10 years history of alcohol abuse	Low plasma testosterone accompanied by a low LH and FSH	[66]
8 men	21.4 ± 4.8 years	Healthy and trained	60 mL of vodka across 3 h period (12 ± 2 drinks consumed) every 30 min and 1 h post-exercise	Pro-apoptotic effects in skeletal muscle following exercise	[67]
4 men and 1 woman	49.2 ± 11.4	Alcoholic cirrhosis	?	Skeletal muscle autophagy	[8]

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
