# Peer review of "Probiotics as Potential Therapeutic Agents: Safeguarding Skeletal Muscle against Alcohol-Induced Damage through the Gut–Liver–Muscle Axis"

_biomedicines, 2024, doi:10.3390/biomedicines12020382_

Round 1

Reviewer 1 Report

Comments and Suggestions for Authors

In the present review, the authors described some of the main effects of alcohol on the skeletal muscle, with emphasis on the gut-liver-muscle axis, which seems to be the main cause of skeletal muscle dysfunction related to the onset of alcoholic liver disease. In general, the review is well-written. Please, see the suggestions and criticisms as follow.

1)    Title. It has to refect the main message or the main findings of the manuscript, to be specific and punctual.

2)    Endocrine abnormality. Because there is great effect on testosterone production, discussion about the differential ethanol effects in men and women is welcome.

3)    Myosteatosis. Discussion about the mechanisms involved in myosteatosis and about the differences between pathological (e.g., insulin resistance state) and physiological (e.g., trained athletes) lipid accumulation in skeletal muscle is welcome.

4)    Figure 2. It is hard to understand. It has to be improved in an easy way. It is not possible to know what the final effects of the ethanol are on skeletal muscle mass, metabolism, and/or function.

5)    Topic 4. A figure summarizing the gut-liver-muscle is welcome. Since this axis is the most important proposition of the manuscript, authors have to summarize their proposition in this figure.

6)    General. Since the alcoholic liver disease is the main focus of the review, authors have to include a related topic. Discussion about the interaction of gut and liver, as well as the liver and muscle, is required.

7)    Topic 5. Proposition of a related figure is welcome.

8)    Table 2. It is not cited into text. Studies investigating the effects of probiotics in liver, and muscle damage induced by ethanol have to be included in this table.

9)    Conclusion. Since the review has not a clear conclusion, it is better to change “Conclusions” by “Final Considerations”. In this topic, authors have to include the perspectives and directions for further studies.

10) Minor comments. A) Define all abbreviations at the first time into the text. B) Page 3, line 131: Please, correct the BMI unit. C) Page 3, lines 133-135: Start the phrases with capital letters.

Comments on the Quality of English Language

Minor editing of English language required.

Author Response

Response to Reviewers

Reviewer 1

Comment 1: Title. It has to refect the main message or the main findings of the manuscript, to be specific and punctual.

Response 1: Thank you so much for the suggestion. We have changed the title in “Probiotics as potential therapeutic agents: safeguarding skeletal muscle aginst alcohol-induced damage through the Gut-Liver-Muscle axis”.

Comment 2: Endocrine abnormality. Because there is great effect on testosterone production, discussion about the differential ethanol effects in men and women is welcome.

Response 2: Thank you very much for the suggestion; we have incorporated the text according to the reviewer's suggestion. With the addition of a paragraph on the “gut-liver-muscle axis”, paragraph 3.3 “Endocrine dysfunctions” has been repositioned and is now 4.3.  Furthermore, within  the same paragraph, a section on the role of myostatin and its levels in individuals with ALD has been included to in support the hypothesis of the  “liver-muscle axis”.

Comment 3: Myosteatosis. Discussion about the mechanisms involved in myosteatosis and about the differences between pathological (e.g., insulin resistance state) and physiological (e.g., trained athletes) lipid accumulation in skeletal muscle is welcome.

Response 3: Thanks so much for the suggestion. We have added into the text the section relating to the physiological accumulation of lipids within the skeletal muscle, emphasizing its primary function as an intracellular energy reserve. Furthermore, the section describing the three fat deposition has been moved up to enhance the flow of the discussion. As a result, paragraph 3.3. of “Myosteatosis” is now labeled as 4.4.

Comment 4: Figure 2. It is hard to understand. It has to be improved in an easy way. It is not possible to know what the final effects of the ethanol are on skeletal muscle mass, metabolism, and/or function.

Response 4: Thank you for the suggestion. We have modified the Figure 2, which is now referred to as Figure 3 according with your request. We hope it is now clearer.

Comment 5: Topic 4. A figure summarizing the gut-liver-muscle is welcome. Since this axis is the most important proposition of the manuscript, authors have to summarize their proposition in this figure.

Response 5: Thanks for the suggestion. In compliance with the reviewers' request, we have included a figure summarizing the “gut-liver-muscle axis” indicated as Figure 2. It is located at the end of the entire paragraph dedicated to this topic.

Comment 6: General. Since the alcoholic liver disease is the main focus of the review, authors have to include a related topic. Discussion about the interaction of gut and liver, as well as the liver and muscle, is required.

Response 6: Thank you very much for the suggestion. We have added a discussion on the gut-liver axis in chapter 2, entitled "Gut microbiota and hypothesis of “gut-liver-muscle axis” in ALD”. In general, the interconnection relationships between the intestine and the liver were described, and various studies were presented on the effects of ethanol on the integrity of the intestinal barrier,  a prominent aspect of the “gut-liver axis”. At the end of chapter 2, we explored how liver alterations can have consequences at the level of skeletal muscle, defining a new axis that connects gut, liver and muscle.

Comment 7: Topic 5. Proposition of a related figure is welcome.

Response 7: Thank you very much. We added the Figure 4 into chapter 6, entitled “Modulation of the gut microbiota – Benefits of probiotics”.

Comment 8: Table 2. It is not cited into text. Studies investigating the effects of probiotics in liver, and muscle damage induced by ethanol have to be included in this table.

Response 8: Thank you for the suggestion. We have incorporated studies on the effects of probiotics on the liver into Table 2, as requested by the reviewers. Unfortunately, we are unable to add studies related to the effects of probiotics on muscle damaged by ethanol to the aforementioned table as no relevant studies were found in the literature.

Comment 9: Conclusion. Since the review has not a clear conclusion, it is better to change “Conclusions” by “Final Considerations”. In this topic, authors have to include the perspectives and directions for further studies.

Response 9: We appreciate the reviewer’s suggestion. We have revised the title of the topic and added guidance and perspectives for future studies.

Comment 10: Minor comments. A) Define all abbreviations at the first time into the text. B) Page 3, line 131: Please, correct the BMI unit. C) Page 3, lines 133-135: Start the phrases with capital letters.

Response 10: Thank you for the suggestions. We have made the following corrections: A) all abbreviations; b) the BMI unit; and c) the capitalization of the phrases at lines 133-135.

Reviewer 2 Report

Comments and Suggestions for Authors

Dear Author,

I have carefully reviewed your manuscript titled "Probiotics treatment can alleviate ethanol-induced damage" and I appreciate the efforts you have put into this study. The article provides an overview of the potential benefits of probiotics in mitigating the harmful effects of alcohol on skeletal muscle. While the topic is interesting and relevant, there are several areas that need improvement and further investigation. In this review, I will provide detailed comments on the following key points:

1. Lack of support from large-scale clinical trials:

The conclusions of this study lack support from large-scale clinical trials. Although there are some studies supporting the potential benefits of probiotics in alleviating alcohol-induced damage to skeletal muscle, more extensive clinical trials are needed to validate the effectiveness and safety of these findings. It is important to conduct well-designed, randomized, controlled trials involving a significant number of participants to obtain robust evidence.

2. Insufficient exploration of the mechanisms of probiotic action:

While the article mentions that probiotics may counteract the effects of alcohol on skeletal muscle by regulating the gut microbiota, increasing muscle glycogen levels, and reducing oxidative stress, there is a lack of in-depth exploration of the specific molecular mechanisms underlying the action of probiotics. Further research is required to elucidate the precise mechanisms through which probiotics exert their effects on skeletal muscle in the context of alcohol consumption.

3. Lack of investigation on different populations:

The study lacks research on the effects of probiotics in different populations. Probiotics may have different effects on individuals of different ages, genders, and health conditions. Therefore, more studies are needed to understand the role of probiotics in different population groups. It would be valuable to investigate whether the benefits observed in one population can be generalized to others.

4. Inadequate comprehensive evaluation of potential side effects and safety of probiotic treatment:

The study lacks a comprehensive evaluation of potential side effects and the safety of probiotic treatment. The use of probiotics can potentially lead to adverse reactions, and it is crucial to assess their safety and potential side effects thoroughly. Further research should focus on investigating the long-term effects and safety profile of probiotic interventions in the context of alcohol-induced damage.

In conclusion, while the review suggests that probiotic treatment can alleviate ethanol-induced damage, it currently lacks sufficient experimental data and support from clinical trials. More research is needed to address these potential limitations and gain a comprehensive understanding of the role of probiotics in mitigating alcohol-induced skeletal muscle damage.

Furthermore, the overall structure of the article appears disjointed, with insufficient logical coherence between the content and context. It is recommended to reorganize the ideas and ensure a logical flow throughout the article.

In conclusion, I believe that addressing these points will greatly improve the clarity, scientific rigor, and impact of your manuscript. I look forward to reviewing the revised version.

Comments on the Quality of English Language

The writing of the article is basically in line with the requirements, there are no major problems with grammar and diction, but the content needs to be revised a lot.

Author Response

Response to Reviewers

Reviewer 2

Comment 1: Lack of support from large-scale clinical trials:

The conclusions of this study lack support from large-scale clinical trials. Although there are some studies supporting the potential benefits of probiotics in alleviating alcohol-induced damage to skeletal muscle, more extensive clinical trials are needed to validate the effectiveness and safety of these findings. It is important to conduct well-designed, randomized, controlled trials involving a significant number of participants to obtain robust evidence.

Response 1: Thank you for bringing this to our attention. We concur with this observation.  In the existing scientific literature, there is a limited body of work discussing the benefits of probiotics in mitigating skeletal muscle damage induced by alcohol abuse. Most studies in the literature focus on the deleterious effects of alcohol abuse on skeletal muscle, employing both in vitro and in vivo experiments in various animal and human models. These studies provide a comprehensive overview of the impact of ethanol on the morpho functional characteristics of skeletal muscle. Similarly, numerous studies examine the positive effects of probiotics on various aspects of muscle function, including enhancements in muscle mass and/or strength, glucose homeostasis, and protein synthesis. In this review, we aimed to integrate these two aspects by presenting: 1) studies on the effects of acute and/or chronic ethanol administration on the skeletal muscle of individuals with a history of alcoholism or those suffering from alcoholic liver disease, and 2) studies on the benefits of probiotics on the muscles of both healthy sedentary and trained subjects. Based on the available literature, we hypothesize that probiotics may have a mitigating effect on muscle damage induced by ethanol treatment. We acknowledge the need for well-designed, randomized, and controlled studies with a substantial number of participants to establish robust evidence. These studies should consider not only the diverse populations involved but also potential side effects of probiotics, particularly in long-term treatment.

Comment 2: Insufficient exploration of the mechanisms of probiotic action:

While the article mentions that probiotics may counteract the effects of alcohol on skeletal muscle by regulating the gut microbiota, increasing muscle glycogen levels, and reducing oxidative stress, there is a lack of in-depth exploration of the specific molecular mechanisms underlying the action of probiotics. Further research is required to elucidate the precise mechanisms through which probiotics exert their effects on skeletal muscle in the context of alcohol consumption.

Response 2: Thank you for your response.  We have integrated the paragraph "Mechanisms of actions of probiotics" (paragraph 6.1) into the manuscript, detailing the primary mechanisms of action of probiotics at the intestinal level, as documented in the literature. On the contrary, we encountered challenges integrating information on the mechanisms of action of probiotics at the level of skeletal muscle damaged by ethanol, as there is a limited presence of studies on this topic. Most of the existing studies focus on pathologies affecting the gut, and investigations into the effects of the probiotics on other organs are still ongoing.  In this review, our intention was to provide insights into the potential benefits of probiotics at the skeletal muscle level, considering the intestine-liver-muscle axis. We aimed to demonstrate that probiotics might play a role in improving glycogen levels, reducing of oxidative stress, and enhancing muscle mass and/or strength. Although the actual mechanisms of action of probiotics against various organs are still under investigation preliminary data from the literature regarding the efficacy of probiotics in skeletal muscle are promising. We hope that future research will delve deeper into this area to validate the hypotheses put forth in this review.

Comment 3: Lack of investigation on different populations:

The study lacks research on the effects of probiotics in different populations. Probiotics may have different effects on individuals of different ages, genders, and health conditions. Therefore, more studies are needed to understand the role of probiotics in different population groups. It would be valuable to investigate whether the benefits observed in one population can be generalized to others.

Response 3: Thank you very much for the suggestion, we have integrated the text according to the reviewer's suggestion on paragraph 6 “Modulation of the intestinal gut microbiota – Benefits of probiotics”.

Comment 4: Inadequate comprehensive evaluation of potential side effects and safety of probiotic treatment:

The study lacks a comprehensive evaluation of potential side effects and the safety of probiotic treatment. The use of probiotics can potentially lead to adverse reactions, and it is crucial to assess their safety and potential side effects thoroughly. Further research should focus on investigating the long-term effects and safety profile of probiotic interventions in the context of alcohol-induced damage.

In conclusion, while the review suggests that probiotic treatment can alleviate ethanol-induced damage, it currently lacks sufficient experimental data and support from clinical trials. More research is needed to address these potential limitations and gain a comprehensive understanding of the role of probiotics in mitigating alcohol-induced skeletal muscle damage.

Response 4: Thanks for the suggestion, we have added additional considerations on the general safety of probiotic treatment and their potential effects in the paragraph "Modulation of gut microbiota - Benefits of probiotics". However, a comprehensive evaluation of the potential side effects and safety of probiotic treatment in patients with alcohol-induced muscle damage was not included in this review due to the limited number of studies available on this topic in the literature.  Furthermore, the preliminary data presented in these studies involve n vitro and/or in vivo experiments on a small number of animal models. Therefore, further research is essential to provide a clearer evaluation of these effects.  Given the growing recognition of the significance of the gut microbiota in the onset and development of various pathologies, the potential use of probiotics in ALD is gaining increasing investigative attention. The primary objective of this review is to summarize the main muscular damages of chronic ethanol consumption and how these damages might be mitigated though the use of probiotics. It is important to note that,  despite the promising nature of some studies  (see reference "Green PG, Alvarez P, Levine JD. Probiotics attenuate alcohol-induced muscle mechanical hyperalgesia: Preliminary observations. Mol Pain. 2022 Jan-Dec;18:17448069221075345. doi: 10.1177/17448069221075345. PMID: 35189754; PMCID: PMC8874179”), there is a scarcity of literature on this subject. As a result, only some potential benefits of probiotics were discussed in the review, without a comprehensive assessment of their possible undesirable effects in different populations and health conditions. Nonetheless, to address the known benefits of probiotics at the intestinal level, we have included some considerations and studies in paragraph "Modulation of the intestinal microbiota – Benefits of probiotics". However, it is important to hightlight that further studies are needed to assess the safety and effectiveness of these treatments.

Round 2

Reviewer 1 Report

Comments and Suggestions for Authors

Authors have properly addressed most suggestions and criticisms in the new version of the manuscript. I have no additional main concerns. Only, please, correct to word “against” in the title.

Comments on the Quality of English Language

Minor editing of English language required.

Author Response

Comment 1: Authors have properly addressed most suggestions and criticisms in the new version of the manuscript. I have no additional main concerns. Only, please, correct to word “against” in the title.

Response 1: Thank you so much for the suggestion. We have correct the word “against” in the title.

Reviewer 2 Report

Comments and Suggestions for Authors

Dear Author,

Thank you for your revised manuscript titled "Probiotics as Potential Therapeutic Agents: Safeguarding Skeletal Muscle Against Alcohol-Induced Damage through the Gut-Liver-Muscle Axis." After reviewing your revisions, I am pleased to inform you that your manuscript has been accepted for publication pending minor revisions. Please find below my comments on the specific points raised during the review process.

1. Regarding the first point, we appreciate your clarification and understand your rationale. However, we suggest that you address the limitations of this review in the "Final Considerations" section of the manuscript to provide a comprehensive overview of the study's scope.

2. Concerning the second point, we have noticed your addition of information on the mechanisms of actions of probiotics in paragraph 6.1. We also understand the limitations of existing research in this field. Overall, we find your response reasonable and acceptable.

3. In response to the third point, we have observed the integration of information on the modulation of the intestinal gut microbiota and the benefits of probiotics in paragraph 6 of your revised manuscript. We believe that the integration is appropriate and can be accepted.

4. Regarding the fourth point, we acknowledge the additional considerations you have included in paragraph 6 regarding the general safety and potential implications of probiotic treatment. As the literature on probiotic treatment is limited, and a comprehensive evaluation of the potential side effects and safety of probiotics was not within the scope of this review, we understand this omission. Overall, we find your explanation on this matter reasonable and acceptable.

Please address the above suggestions in your final manuscript. Once these minor revisions have been made, your manuscript will be ready for publication. Thank you for your efforts in revising the manuscript, and we appreciate your valuable contribution to the field.

Comments on the Quality of English Language

Dear Editor,

I have carefully reviewed the revised manuscript titled "Probiotics as Potential Therapeutic Agents: Safeguarding Skeletal Muscle Against Alcohol-Induced Damage through the Gut-Liver-Muscle Axis." I am pleased to inform you that the manuscript is now suitable for publication pending minor revisions. Please find below my comments on the specific points raised during the review process.

1. Regarding the first point, I understand the author's perspective, and it would be beneficial to address the limitations of this review in the "Final Considerations" section of the manuscript. This addition will provide a clear description of the scope and boundaries of the present review.

2. Concerning the second point, I have noticed that the author has supplemented information on the mechanisms of actions of probiotics in paragraph 6.1. Considering the limitations of the existing research in this field, I find the author's response reasonable and acceptable.

3. Regarding the third point, I have observed the integration of information on the modulation of the intestinal gut microbiota and the benefits of probiotics in paragraph 6 of the revised manuscript. The integration appears logical and can be accepted.

4. Concerning the fourth point, I have noted that the author has addressed the general safety and potential implications of probiotic treatment in paragraph 6 of the "Modulation of the intestinal gut microbiota Benefits of probiotics" section. Given the limited research on the potential side effects and safety of probiotic treatment in the literature, it is understandable that a comprehensive evaluation was not included in this review. Overall, I find the author's explanation on this matter reasonable and acceptable.

Once the minor revisions based on the above suggestions have been made, the manuscript will be ready for publication. I commend the author for their efforts in revising the manuscript, and I believe that this study will make a valuable contribution to the field.

Author Response

Comment 1: Regarding the first point, we appreciate your clarification and understand your rationale. However, we suggest that you address the limitations of this review in the "Final Considerations" section of the manuscript to provide a comprehensive overview of the study's scope.

Response 1: Thank you very much for your suggestion. We have added the limitations of this review at the end of paragraph 8 entitled "Final Considerations".